# Evaluation of Genetic Parameters of Growth of Pelibuey and Blackbelly Sheep through Pedigree in Mexico

**DOI:** 10.3390/ani12060691

**Published:** 2022-03-10

**Authors:** Víctor Rodríguez Hernandez, Vicente Vega Murillo, Roberto Germano Costa, Conrado Parraguirre Lezama, Maria De Los Angeles Valencia de Ita, Omar Romero-Arenas

**Affiliations:** 1Colegio de Posgraduados, Campus Puebla, Puebla 72860, Mexico; rodriguezjv@colpos.mx; 2INIFAP, Centro de Investigación Regional Golfo-Centro, Veracruz 94277, Mexico; vega.vicente@live.com.mx; 3Programa de Doutorado Integrado em Zootecnia, Universidade Federal da Paraíba, João Pessoa 58051-900, Brazil; rcqueiroga@uol.com.br; 4Centro de Agroecologia, Instituto de Ciencias, Benemérita Universidad Autónoma de Puebla, Edificio VAL 1, Km 1,7 Carretera a San Baltazar Tetela, San Pedro Zacachimalpa, Puebla 72960, Mexico; conrado.parraguirre@correo.buap.mx (C.P.L.); maria.valenciadeita@correo.buap.mx (M.D.L.A.V.d.I.)

**Keywords:** pedigree, sheep, genetic effect, direct heritability

## Abstract

**Simple Summary:**

The importance of sheep production in Mexico has been increasing in recent years. Animal performance can be improved through continuous selection and cumulative genetic management. Studies on non-genetic factors that influence the growth characteristics of sheep, due to their positive genetic correlation with other live weights, can present data of great interest to small producers with limited resources. In the present study, genealogical and functional information from the historical archive of the Mexican Association of Sheep Breeders (AMCO) was used. The objective was to estimate the heritability and genetic correlations of the growth of ewes born and weaned at different times of the same year from different herds with pedigree registration. The expected results could be incorporated into genetic selection programs with a favorable impact on the local economy of small sheep producers in Mexico.

**Abstract:**

Birth weight (BW) and weaning weight (WW) data from Pelibuey and Blackbelly lambs belonging to the Asociación Mexicana de Criadores de Ovinos (AMCO) were used with the objective of estimating genetic parameters (heritability and genetic correlations) and analyzing the growth characteristics of ewes born and weaned at different times of the same year from different herds with pedigree registration. In the case of Pelibuey lambs, the animal model included the weaning weight at 75 days of age, considering the direct additive genetic effect, maternal additive genetic effect, covariance between direct and maternal effects, as well as the permanent environmental effect of the mother. The direct estimators of heritability for Pelibuey were BW = 0.01 ± 0.021 and WW = 0.31 ± 0.074 and for Blackbelly they were BW = 0.05 ± 0.042 and WW = 0.41 ± 0.146. In the case of the maternal heritability estimators in Pelibuey they were BW = 0.02 ± 0.040 and WW = 0.21 ± 0.121 and for Blackbelly they were BW = 0.12 ± 0.054 and WW = 0.28 ± 0.121. The magnitude of the estimates of genetic correlations between direct and maternal effects for adjusted weaning weight at 75 days of age indicate that genetic progress may be slow in a breeding program. However, these selection results could be included in the short term in the breeding programs for the Pelibuey and Blackbelly breeds in Mexico, for livestock development in low-income rural areas.

## 1. Introduction

The sheep (*Ovis aries*) is one of the earliest domesticated livestock species whose ancestors were primarily distributed in the fertile crescent approximately 10,000 years ago [1]. Livestock production systems in tropical countries have successfully incorporated hair sheep breeds; in addition, it is estimated that the world sheep population comprises approximately 10% of production systems [2]. Sheep have various adaptive mechanisms such as their fur that makes them uniquely qualified to be productive in hot and semi-warm environments, humid environments, as well as in environments with scarce resources [3,4].

Pelibuey and Blackbelly breeds were the first hair breeds introduced in Mexico and currently constitute the basis of tropical sheep production in the country [5]. However, they present a high adaptability, which has led to sheep production being widespread throughout the Mexican territory [6]. Both breeds are raised and managed under the same conditions and share important characteristics such as the absence of seasonality, high fertility and prolificacy rates, high adaptability to heat, humidity, parasites, food scarcity, among other adverse environmental conditions, which are key characteristics to achieve social and productive sustainability [7].

There is an urgent need to increase productivity to improve the income of small sheep producers and meet the demand for livestock products from the growing human population [8]. The efficiency and profitability of animal production are determined by the genetic merit of the animals, to increase the efficiency in the use of nutrients on growth [4].

Genomic selection (GS) aims to increase precision and decrease generation intervals, obtaining genetic gains from young animals, allowing their early selection with a potentially favorable effect on reproduction [9]. In the case of sheep, it has opened the perspective of genomic selection (GS) in dairy, meat, and wool products [10], improving and increasing profits at low costs.

Sheep producers of the Pelibuey and Blackbelly breeds in Mexico need to implement selection systems based on productive records with characteristics of economic importance, which allows them to select their sheep based on estimates of the animal’s genetic value, especially recommended for those characteristics with low heritability [11]. Genetic evaluations analyze genealogical information and productive behavior in two phases: (1) estimation of variance components and calculation of genetic parameters; and (2) prediction of breeding values and their expected presentation in the progeny. Genetic parameters such as heritability and genetic correlation characterize populations according to genetic influences and allow timely selection programs to be defined [12]. The prediction of breeding values provides objective tools for the identification of genetically superior individuals and their selection as future breeders, guaranteeing better productive characteristics. Genetic progress in selection schemes depends on the use of correct models for genetic evaluation. Models are simplifications of reality and are never completely perfect, so tools are needed to analyze systematic errors. Bias is the difference between estimated breeding values and true breeding values, which could lead to overestimation or underestimation of genetic bias and poor selection decisions (e.g., selecting too many young individuals instead of keeping old individuals). Likewise, values of the slope of the regression and actual breeding values over the estimated breeding values of less than 1 could imply an overdispersion and give rise to an overestimation of the genetic merit of the preselected candidates. On the other hand, an unbiased estimate of precision (the correlation between actual breeding values and estimated breeding values) is important to correctly predict the response to selection [13].

## 2. Materials and Methods

### 2.1. Data

Genealogical and functional information used in this study belongs to the historical archives of the Mexican Association of Sheep Breeders (AMCO). A total of 285 Pelibuey rams and 114 Blackbelly rams were used to produce 8276 and 3890 evaluated lambs, respectively. The number of breeding ewes was 2910 for the Pelibuey breed and 1196 for the Blackbelly breed, belonging to 87 herds for Pelibuey and 52 for Blackbelly, covering the birth years from 2014 to 2020; only the weights at birth of lambs born alive and surviving at weaning were considered. Individual weaning weight records were adjusted at 75 days of age (WW), using the following Equation (1):WW = [(weaning weight − birth weight)/age in days at weaning] × 75 + birth weight(1)

### 2.2. Statistical Models

The animal model for birth weight in both Pelibuey and Blackbelly (Equation (2)) included the direct additive genetic effect, the maternal additive genetic effect, and the maternal permanent environment effect, but did not include covariance between direct and maternal genetic effects.
y = Xβ + Z_a_a + Z_m_m + Z_p_P + ewithout covariance (a, m) = 0(2)

The animal model for weight adjusted to 75 days of age (Equation (3)) included the direct additive genetic effect, the maternal additive genetic effect, and the covariance between the direct and maternal genetic effects. However, it did not include the effect of the mother’s permanent environment. Equation (4) considered the same effects as Equation (3), where the effect of the mother’s permanent environment was included.
y = Xβ + Z_a_a + Z_m_m + ewithout covariance (a, m) = Aσam(3)
y = Xβ + Z_a_a + Z_m_m + Z_p_P + ewith covariance (a, m) = Aσam(4)
where, in Equations (2)–(4):y = the vector of records of birth weight or weaning weight.β = a vector of fixed effects (sex of the lamb, type of lambing, contemporary group, and age of the ewe).a = an unknown direct additive genetic random effects vector.m = an unknown maternal additive genetic random effects vector.p = an unknown vector of random effects of the permanent maternal environment.e = an unknown vector of random effects of the temporal environment.X = Z_a_, Z_m_, and Z_p_ are known incidence matrices that relate the records to β, a, m, and p, respectively.

As fixed effects, all animal models included the sex of the lamb, the type of lambing or weaning (single, double, triple, quadruple), the contemporary group, and the age of the mother at lambing as a linear covariate. The working hypothesis assumed that the direct additive genetic effects, maternal additive genetic effects, maternal permanent environmental effects, and residual effects were normally distributed with a mean of 0 and a structure of variances and covariances that depended on the assumptions of each model. The variance and covariance (Equation (5)) for the random effects of the animal model was:(5)$$V\left[\begin{array}{l} a\\ m\\ p\\ e \end{array}\right]=\left[\begin{array}{cccc} A\sigma_{a}^{2} & A\sigma_{am} & 0 & 0\\ A\sigma_{am} & A\sigma_{m}^{2} & 0 & 0\\ 0 & 0 & I_{N_{d}}\sigma_{p}^{2} & 0\\ 0 & 0 & 0 & I_{N}\sigma_{e}^{2} \end{array}\right]$$
where:*A* = the Wright matrix of additive kinships between all animals in the pedigree:$\sigma_{a}^{2}$ = the direct additive genetic variance.$\sigma_{m}^{2}$ = the maternal additive genetic variance.$\sigma_{am}$ = the genetic covariance between direct and maternal effects.$\sigma_{p}^{2}$ = the variance of the permanent maternal environment.$\sigma_{e}^{2}$ = the variance of the temporal environment or residual variance.$I_{N_{d}}$ = an identity matrix of equal size to the number of ewes.*I_N_* = an identity matrix of equal size to the total number of observations.

### 2.3. Genetic Parameter Estimators

Heritability estimators were calculated from the estimators of the variance components. Heritability was calculated as the proportion of phenotypic variance due to additive genetic variance. The standard errors of the heritability estimators were approximate and calculated using the average information matrix [14] and the Delta Method [15].

Estimators were obtained for phenotypic variance ($\sigma_{p}^{2}$ = $\sigma_{a}^{2}$ + $\sigma_{m}^{2}$ + $\sigma_{am}$ + $\sigma_{pe}^{2}$ + $\sigma_{e}^{2}$), heritability for direct additive genetic effects ($h_{a}^{2}$ = $\sigma_{a}^{2}$/$\sigma_{p}^{2}$), heritability for maternal additive genetic effects ($h_{m}^{2}$ = $\sigma_{m}^{2}$/$\sigma_{p}^{2}$), correlation between direct and maternal additive genetic effects ($r_{am}$ = $\sigma_{am}$/($\sigma_{a}^{2}$$\sigma_{m}^{2}$)^1/2^), fraction of the phenotypic variance due to effects of the permanent maternal environment ($c^{2}$ = $\sigma_{pe}^{2}$/$\sigma_{p}^{2}$) and residual variance as a proportion of the phenotypic variance ($e^{2}$ = $\sigma_{e}^{2}$/$\sigma_{p}^{2}$).

The components of variance and genetic parameters were estimated with an animal model and Maximum Restricted Likelihood Free of Derivatives, using the MTDFREML program [16]. Analyses were performed independently for each characteristic within each breed. Heritability estimators were calculated from the variance component estimator criteria [−2 log(L)] < 1 × 10^−9^ (where L represents the likelihood function) [17].

## 3. Results

Table 1 shows the analyzed datasets. The number of records (observations) analyzed in the Pelibuey breed was approximately twice as high as in the Blackbelly breed. The mean birth weight (BW) and weaning weight (WW) adjusted to 75 days of age were around 3 and 18 kg in both breeds, respectively.

On the other hand, for the birth weight in the Pelibuey breed, the maternal additive genetic variance was approximately two times greater than the direct additive genetic variance (Table 2).

However, the most important component in determining birth weight was the effect of the permanent maternal environment, since the variance of that effect was 0.025 kg^2^, while the maternal additive genetic variation was 0.024 kg^2^ for Blackbelly and 0.004 kg^2^ for Pelibuey approximately. Conversely, for weaning weight adjusted to 75 days of age in the Pelibuey breed, the direct additive genetic variance was greater than the maternal additive genetic variance (2.0 vs. 1.4 kg^2^, respectively). Furthermore, the variance of the permanent maternal environment (0.90 kg^2^) was lower than the direct and maternal additive genetic variances.

For the birth weight in the Blackbelly breed, the maternal additive genetic variance was approximately two times greater than the direct additive genetic variance (0.011 vs. 0.024 kg^2^), as was the case for birth weight in the Pelibuey breed. Although they were estimated with different models, the direct and maternal additive genetic variances for weaning weight adjusted to 75 days of age in the Blackbelly breed were similar to the direct and maternal additive genetic variances for weaning weight adjusted to 75 days of age in the Pelibuey breed.

The phenotypic variances for birth weight were similar, with values of 0.24 kg^2^ and 0.20 kg^2^ for Pelibuey and Blackbelly, respectively. The phenotypic variances for weaning weight adjusted to 75 days of age were different (6.63 vs. 5.36 kg^2^), which may have been due to the model, as it did not include the effect of the permanent maternal environment on the Blackbelly breed. The estimators of the genetic parameters for the growth characteristics evaluated in Pelibuey and Blackbelly are shown in Table 3.

Direct heritability estimators for birth weight were close to zero in both breeds (0.01 and 0.05 for Pelibuey and Blackbelly, respectively). The maternal heritability estimator for birth weight in the Blackbelly breed was higher than the maternal heritability estimator for birth weight in the Pelibuey breed (0.12 vs. 0.02); however, both heritability estimators were low, indicating that maternal additive genetic effects for birth weight are poorly heritable. For birth weight in Pelibuey, the variance of the permanent maternal environment with the proportion of phenotypic variance was 10 times greater than direct and maternal heritabilities (11 vs. 1 and 2%).

For birth weight in Blackbelly, the variance of the permanent maternal environment with proportion of phenotypic variance and direct heritability were similar (4 vs. 5%, respectively), contrary to what happened for birth weight in the Pelibuey breed. On the other hand, maternal heritability for birth weight in Blackbelly was 12% higher than direct heritability and variance of the permanent maternal environment as a proportion of phenotypic variance. The direct additive genetic effects for weaning weight adjusted to 75 days of age were moderately heritable in the Pelibuey (31%) and Blackbelly (41%) breeds, indicating that the selection of outstanding parents based on their expected differences in progeny (genetic values) allows for the increase of weaning weights.

Maternal genetic effects for weight adjusted at 75 days of age had heritability of about 25% in Pelibuey and Blackbelly, being less heritable than direct genetic effects. However, direct and maternal genetic effects for weaning weight adjusted to 75 days of age were highly and negatively correlated in both races, and were more strongly correlated in the Pelibuey breed than in the Blackbelly breed (−0.73 vs. −0.66, respectively).

## 4. Discussion

In both the Pelibuey and Blackbelly breeds, genetic covariances between direct and maternal effects for adjusted weaning weight at 75 days of age were negative, indicating that offspring growth is antagonistic to milk production of the sheep, this means that, if we select for greater growth, we would demerit the milk production (maternal capacity) of the sheep in both breeds [18]. The results are similar to those obtained by Maria et al. [19] with Romanov sheep. Another study reported by Analla and Sedarrilla [20] reached the same results with Merino sheep, where they concluded that these characteristics could be taken towards the production of quality females in Spain, with larger litter sizes and better maternal abilities.

Cloete et al. [21] presented lower milk production with Dohne-Merino sheep, like the authors Tosh and Kemp [22] with Romanov sheep; however, Pitono and James [23], with tropical sheep breeds (Boujenane and Kansari), obtained a higher weight, like the authors Boujenane and Kansari [24], where they obtained covariance values −0.55 with Timahdite sheep and values lower than those reported in the present investigation due to direct maternal effects on body weight.

In this context, Simm et al. [25] reported the estimation of heritability by direct additive genetic effects, 0.054 for birth weight (BWT) and 0.177 for weaning weight (WWT) with Suffolk ewes, indicating that the direct response to selection by birth weight would be slow in a breeding program, values similar to those estimated in the present study.

Previous studies with Merino, Romanov, Welsh Mountain, Targhee, and Suffolk sheep breeds have found maternal heritability estimators like those in the present study [19,20,26,27,28]. In contrast, for the Swedish fine-wool breeds, Baluchi, Dala, Coopworth, Chios, Polypay, Dorper, and Columbia, lower values have been reported [29,30,31,32,33,34,35,36].

In contrast, the direct heritability estimators reported by Lewer et al. [37] for Merino sheep from Western Australia was WW = 0.32–0.39, similar results to those found for the Pelibuey breed, but lower than those found for the Blackbelly breed in the present investigation. Vaez-Torshizi et al. [38] mention that the weight at birth for Australian Merino sheep would be more feasible and accelerated by seeing the effect of genetic correlations between direct and maternal additive effects ($r_{am}$ = −0.59), lower results than those of the present investigation. In addition, the scientific literature indicates that the magnitude of the estimators of the genetic correlation between direct and maternal effects is highly variable, finding values of −0.98 [19], −0.42 [23], −0.78 [34], −0.60 [38], −0.90 [39], and −0.53 [40].

Delphino-Medrado et al. [29] performed a meta-analysis of diversified studies of genetic parameters in sheep, obtaining heritability parameters between 0.121 and 0.39 from a total of 191 articles, using a random model of reliable estimates. ranges similar to those found in the present study. However, Qiao et al. [41], showed that genetic variation in birth weight is predominantly explained by fetal genetics rather than maternal genetic sources of variation. As in the present work, other authors [31,33,42,43] have found in different sheep breeds (Dorper, Merino, Polled Dorset, Baluchi) that birth weight is a poorly heritable characteristic. Several studies have confirmed that maternal genetic effects for birth weight are lesser heritable, for example in the Harnali breed (0.23 [44]) in the Egyptian Barki sheep breed (0.07 [32]), in the Dala breed (0.42 [33]), and in the Suffolk breed of sheep (0.54 [25]). McGlothlin [45], working with simulated data, concluded that “estimators of the genetic correlation between direct and maternal effects may be negative due not only to genetic antagonism, but also due to additional variation between rams or between rams × year”; however, such effects are generally not considered in the statistical model for information analysis.

Matebesi-Rantimo et al. [26] indicate that genetic selection schemes for initial weight and carcass characteristics should use models that consider more than just direct additive effects, failure to do so will lead to overestimation of genetic progress and increase the chances of making decisions.

The scientific literature reports the importance of direct heritability factors for weaning weight, in this sense, Analla and Serradilla [20] report results of a study carried out in Spain with the Merino breed, in which 126 rams and 964 sheep were used and 3355 records were analyzed, where they obtained a direct heritability estimator of 0.09 for weaning weight at 60 days of age, lower values compared to the ones estimated in the present study. In a study conducted within the US National Sheep Improvement Program, they found that additive maternal effects on weaning weight were positively associated with litter size in Suffolk and Polypay breeds [27]. However, the scientific literature has also reported the importance of direct heritability factors for weaning weight, in that way, Analla and Serradilla [20] reported results of a study carried out in Spain with the Merino breed, in which the authors analyzed the weaning weight to 30, 60, and 90 days of age of 4425 lambs and 3355 litters at delivery of 964 ewes, obtaining a direct heritability estimator of 0.06, 0.09, and 0.14 for the weaning weight at 30, 60, and 90 days of age, respectively, values lower than those estimated in the present study. In this sense, the parameter of weight at weaning (WW) estimated in the present study, can be used in sheep breeding programs, especially in developing and underdeveloped countries, where the breeding of Pelibuey and Blackbelly sheep is an important livestock activity, as in the case of Mexico.

The estimators of direct heritability for weaning weight obtained in the present study may differ from those reported in the literature due to differences in breed, management, environment, number of records analyzed, intensity of selection, etc.

Al-Shorepy and Notter [42] reported the estimates of the fraction of phenotypic variance due to permanent maternal environmental effects, obtaining higher values compared to the estimation of heritability by direct additive genetic effects and heritability by maternal additive genetic effects (0.31, 12, and 0.07). Results were similar to the values estimated in the present study for the Pelibuey breed (0.11, 0.01, and 0.02).

In the case of the Blackbelly breed, the results obtained in the present investigation were lower (0.04, 0.05, and 0.12). However, Tosh and Kemp [22] obtained results similar to the values estimated in the present study, the authors conclude that the effects of the permanent maternal environment do not influence the birth weight of the lambs.

Vaez-Torshizi et al. [38] report that the early (indirect) selection for body weight at weaning or 10 months will achieve a substantial proportion (between 53 and 81%) of direct response for performance at later ages (16 and 22 months). In this sense, the direct additive genetic effects for adjusted weaning weight at 75 days of age were moderately heritable in the Pelibuey (31%) and Blackbelly (41%) breeds, which allows weaning weights to be increased. Hence, further studies with a larger data set, and probably using other approaches, should be carried out to confirm the parameters’ estimates obtained and the deductions outlined here.

## 5. Conclusions

Direct and maternal heritability estimates for weight at 75 days of age for both Pelibuey and Blackbelly breeds were significantly higher than direct and maternal heritability estimates for birth weight. For adjusted weaning weight at 75 days of age, direct genetic effects were more important than maternal genetic effects, as direct heritability estimates were larger than maternal heritability estimates. The magnitude of the heritability estimators indicate that it is very feasible to improve (increase) the weight at 75 days of age through a selection program. This selection program could be based on expected differences in progeny or breeding values, as genetic evaluations of the Pelibuey and Blackbelly breeds are currently underway in Mexico.

The magnitude of the estimates of genetic correlations between direct and maternal effects for adjusted weaning weight at 75 days of age indicate that genetic progress may be slow in a breeding program; however, this magnitude could be due to the influence on the data of certain effects not considered in the model (e.g., boar × herd). Finally, the differences between the genetic parameter estimators obtained in this study and some estimators published in the scientific literature could be due to differences in management, number of observations, breed, environment, and selection intensity, among other factors.

## Figures and Tables

**Table 1 animals-12-00691-t001:** Descriptive statistics for birth weight (BW) and weaning weight (WW) adjusted to 75 days of age of Pelibuey and Blackbelly lambs.

Descriptive Statistics	Pelibuey		Blackbelly	
	(BW)	(WW)	(BW)	(WW)
Number of observations	5579	4418	2791	2226
Mean	3.05	17.99	2.97	17.09
Minimum value	1.10	7.24	1.2	7.70
Maximum value	5.50	31.16	4.9	29.47
Standard deviation	0.63	3.70	0.59	3.32
Coefficient of variation (%)	20.77	20.57	19.83	19.45
Information Structure				
Number of rams	285	267	114	112
Number of ewes	2910	2482	1196	1088
Number of animals in the pedigree	8276	8276	3890	3890
Number of herds	87	83	52	52
Number of contemporary groups	276	264	148	147

**Table 2 animals-12-00691-t002:** Estimators of variance components for birth weight (BW) and weaning weight adjusted to 75 days of age (WW) of Pelibuey and Blackbelly lambs.

	Variance Component ^a^					
Pelibuey	$\sigma_{a}^{2}$	$\sigma_{m}^{2}$	$\sigma_{am}$	$\sigma_{pe}^{2}$	$\sigma_{e}^{2}$	$\sigma_{p}^{2}$
(BW)	0.00189	0.00441	-	0.025487	0.21076	0.24254
(WW)	2.03485	1.40724	−1.23509	0.903515	3.52295	6.63346
Blackbelly						
(BW)	0.01125	0.02432	-	0.008095	0.16499	0.20865
(WW)	2.20411	1.48457	−1.19777	-	2.87047	5.36137

^a^ = $\sigma_{a}^{2}$
is the direct additive genetic variance; $\sigma_{m}^{2}$
is the maternal additive genetic variance; $\sigma_{am}$
is the covariance between direct and maternal genetic effects; $\sigma_{pe}^{2}$
is the variance of the permanent maternal environment; $\sigma_{e}^{2}$
is the variance of the error; $\sigma_{p}^{2}$
is the phenotypic variance.

**Table 3 animals-12-00691-t003:** Estimators of genetic parameters and their standard errors for birth weight (BW) and weaning weight adjusted to 75 days of age (WW) of Pelibuey and Blackbelly lambs.

	Genetic Parameter ^a,^*				
Pelibuey	$h_{a}^{2}$	$h_{m}^{2}$	$r_{am}$	$c^{2}$	$e^{2}$
(BW)	0.01 ± 0.021	0.02 ± 0.040	-	0.11 ± 0.043	0.87 ± 0.024
(WW)	0.31 ± 0.074	0.21 ± 0.119	−0.73 ± 0.556	0.14 ± 0.074	0.53 ± 0.055
Blackbelly					
(BW)	0.05 ± 0.042	0.12 ± 0.054	-	0.04 ± 0.054	0.79 ± 0.038
(WW)	0.41 ± 0.146	0.28 ± 0.121	−0.66 ± 0.640	-	0.54 ± 0.106

^a,^* = $h_{a}^{2}$
is heritability for direct additive genetic effects; $h_{m}^{2}$
is heritability for maternal additive genetic effects; $r_{am}$
is correlation between direct and maternal additive genetic effects; $c^{2}$
is fraction of the phenotypic variance due to effects of the permanent maternal environment; $e^{2}$
is residual variance as a proportion of the phenotypic variance.

## Data Availability

Informed consent was obtained from all subjects involved in the study.

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
