# Peer review of "Evaluation of Genetic Parameters of Growth of Pelibuey and Blackbelly Sheep through Pedigree in Mexico"

_animals, 2022, doi:10.3390/ani12060691_

Round 1
Reviewer 1 Report
Dear authors,
The English of the manuscript is really poor, some sentences make no sense; some words are in Spanish and others do not exist in English or they have different meanings. I started to report some errors in the line-by-line comments, but then I gave up because there were too many of them.
Moreover, the manuscript is quite confusing.
Please try to be clearer and to use more appropriate wording.
Another limit of the paper is its limited novelty. I suggest you to include a paragraph in the introduction about the use of genomic selection in livestock focusing on the several studies in sheep (e.g., https://doi.org/10.1017/s1751731115002049; https://doi.org/10.3168/jds.2018-15333; https://doi.org/10.3168/jds.2011-4980; https://doi.org/10.1186/s12711-020-00567-1).
Line-by-line comments.
Line 15: please fix the acronyms for birth weight and weaning weight, they are usually indicated with BW and WW.
Line 17: “Weeding weight” what do you mean?
Line 18: “animal models for a single characteristic” should be “single trait animal models” or something like that.
Line 29: replace the comma with a full stop after covariate.
Line 31: “weeding” please fix it
Lines 31-33: please add standard errors for the estimates.
Lines 72-73: please fix this sentence, “which were born…” seems to refer to Pedigree record (why capital letter?)
Line 78: “stallions” ?? Do you use horses to “produce” lambs? The males for reproduction in sheep are “rams”.
Lines 77-81: too long and confusing sentence.
Line 93: “without cov (a,m)= 0” this makes no sense, please write “without cov(a,m)” or “with cov(a,m)=0”.
Line 91-114: any justifications for all these different models?
Lines 109-111: why “unknown”? It is uncommon to have it written. Since you want to compute a, it is clear that you don’t know the additive genetic effects.
Line 112: “is an unknown vector of random effects of the temporal environment.” It is the vector of random residual, right?
Line 115: what about the parity of the ewe?
Line 116: in the abstract you do not have “quadruple”.
Line 117: “the mother at calving with linear covariate” should be “the mother at calving as linear covariate”.
Line 147: reference for the “MTDFREML” software?
Line 148: “A var convergence criterion” please check this
Line 149-153: this sentence is already reported at lines 134-138
Line 155: please consider to change “shows the information analyzed” with “shows the analyzed datasets”.
Line 160, Table 1: please fix minimum and maximum without accents. Please change stallions with rams. Please fix the acronyms, PN should be BW.
Line 172, Table 2: “Componente de varianza” is not English
Line 191, Table 3: According to the text, this table should show the heritability, right? However, your symbols are the same of Table2.
Lines 222-227: sentence too long and confusing. “with Merino, with Dohne Merino”, etc. Did you mean that those authors analyzed those breeds? Any better wording compared to “with”? Something like “Maria et al. [15] analyzed data from Ronanov, Analla e Sedarilla and reported heritabilities of ….”.
Line 223: Is “Analla y Sedarrilla” a sheep breed? Or “y” must be “and”?
Line 229-234: again, sentence too long and confusing.
Author Response
Reply to reviewer 1
Thanks for the comments, I also comment the following:
The manuscript fits very well with the scope of the special issue of the journal (Animal Genetics and Genomics). The work presented presents results in which the estimators of direct heritability of two breeds of sheep introduced in Mexico, where there are few studies related to the subject, stand out of interest.
Likewise, I inform you that each of your comments were met to substantially improve the work presented. In the attached file, the changes marked in yellow color are shown.
Reviewer Comments
- The English of the manuscript is poor; some sentences make no sense; some words are in Spanish and others do not exist in English or they have different meanings.
Answer
The dramatic quality of the manuscript has improved markedly, and the suggested lines marked in yellow in the text are more clearly detailed.
It is worth mentioning that the work was reviewed by a native speaker of the English language (USA) to comply with this observation. If you think that this answer is not enough, I can tell you that we can opt for the style correction offered by the magazine; This is to comply with the language requirement, and not because of that, the language is an obstacle that prevents the publication of these results.
Reviewer Comments
- I suggest you include a paragraph in the introduction about the use of genomic selection in livestock focusing on the several studies in sheep (e.g., https://doi.org/10.1017/s1751731115002049; https://doi.org/10.3168/jds.2018-15333; https://doi.org/10.3168/jds.2011-4980; https://doi.org/10.1186/s12711-020-00567-1).
Answer
A paragraph was included in the introduction on the use of genomic selection in cattle and the suggested bibliography was attached to the work. Likewise, I appreciate the support of the suggested appointments.
Reviewer Comments
Line-by-line comments.
Line 15: please fix the acronyms for birth weight and weaning weight, they are usually indicated with BW and WW. Answer: Done
Line 17: “Weeding weight” what do you mean? Answer: Dramatic error, the correct thing is weaning weight.
Line 18: “animal models for a single characteristic” should be “single trait animal models” or something like that. Answer: Writing is improved
Line 29: replace the comma with a full stop after covariate. Answer: Done
Line 31: “weeding” please fix it. Answer: Done
Lines 31-33: please add standard errors for the estimates. Answer: Done
Lines 72-73: please fix this sentence, “which were born…” seems to refer to Pedigree record (why capital letter?). Answer: Writing is improved
Line 78: “stallions” ?? Do you use horses to “produce” lambs? The males for reproduction in sheep are “rams”. Answer: The writing is improved and the correct term “rams” is used.
Lines 77-81: too long and confusing sentence. Answer: Writing is improved
Line 93: “without cov (a,m)= 0” this makes no sense, please write “without cov(a,m)” or “with cov(a,m)=0”. Answer: Cov refers to the covariance, the term is completed
Line 91-114: any justifications for all these different models? Answer: According to the literature, they are the most used models to evaluate heritability characteristics in sheep.
Lines 109-111: why “unknown”? It is uncommon to have it written. Since you want to compute a, it is clear that you don’t know the additive genetic effects. Answer: Writing is improved
Line 112: “is an unknown vector of random effects of the temporal environment.” It is the vector of random residual, right? Answer: Yes
Line 115: what about the parity of the ewe? Answer: Writing is improved
Line 116: in the abstract you do not have “quadruple”. Answer: Writing is improved
Line 117: “the mother at calving with linear covariate” should be “the mother at calving as linear covariate”. Answer: Writing is improved
Line 147: reference for the “MTDFREML” software? Answer: Requested reference is attached
Line 148: “A var convergence criterion” please check this. Answer: Writing is improved
Line 149-153: this sentence is already reported at lines 134-138. Answer: Repeated paragraph is deleted
Line 155: please consider to change “shows the information analyzed” with “shows the analyzed datasets”. Answer: Writing is improved
Line 160, Table 1: please fix minimum and maximum without accents. Please change stallions with rams. Please fix the acronyms, PN should be BW. Answer: Writing is improved
Line 172, Table 2: “Componente de varianza” is not English. Answer: Writing is improved
Line 191, Table 3: According to the text, this table should show the heritability, right? However, your symbols are the same of Table2. Answer: Writing is improved, and the error is corrected
Lines 222-227: sentence too long and confusing. “with Merino, with Dohne Merino”, etc. Did you mean that those authors analyzed those breeds? Any better wording compared to “with”? Something like “Maria et al. [15] analyzed data from Ronanov, Analla e Sedarilla and reported heritabilities of ….”. Answer: The discussion of the work is considerably improved.
Line 223: Is “Analla y Sedarrilla” a sheep breed? Or “y” must be “and”? Answer: The discussion of the work is considerably improved.
Line 229-234: again, sentence too long and confusin. Answer: The discussion of the work is considerably improved.

Reviewer 2 Report
The manuscript by Hernandez et al. reports on the genetic effects found for birth weight and weaning among other in Pelibuey and Blackbelly lambs. The analysis in this study has been done by a thorough statistical approach. However, the aim of the analysis is not made fully clear. In particular in the abstract section, I cannot find a proper statement on the purpose of this study. Besides descriptions of methods, the authors should state the hypothesis of this research work and importance of the results for breeding decisions.
The term “genetic values” has been used quite widely in the manuscript, but it not properly defined. It should be clearly stated, which traits are investigated and why they do support breeding.
Table 1: “stallion” is uncommon; use “billy goat”
Table 2: Use English terms only (“Componente de varianza“); see also other tables for language corrections
Author Response
Reply to reviewer 2
Thanks for the comments, I also comment the following:
The manuscript fits very well with the scope of the special issue of the journal (Animal Genetics and Genomics). Likewise, I inform you that each of his comments were addressed to substantially improve the work presented. The attached file shows the changes marked in green.
Reviewer Comments
The manuscript by Hernandez et al. reports on the genetic effects found for birth weight and weaning among other in Pelibuey and Blackbelly lambs. The analysis in this study has been done by a thorough statistical approach. However, the aim of the analysis is not made fully clear. In particular in the abstract section, I cannot find a proper statement on the purpose of this study. Besides descriptions of methods, the authors should state the hypothesis of this research work and importance of the results for breeding decisions.
Answer
The main objective is added in the abstract section, where the purpose of this study is clearer. In addition, the hypothesis of the work is detailed in the methodology section and finally the importance of the results found in discussion and conclusions are highlighted.
Reviewer Comments
The term “genetic values” has been used quite widely in the manuscript, but it not properly defined. It should be clearly stated, which traits are investigated and why they do support breeding.
Answer
The reviewer's suggestion was made and it is better detailed in the text of the document
Reviewer Comments
Table 1: “stallion” is uncommon; use “billy goat”
Answer:
Done
Reviewer Comments
Table 2: Use English terms only (“Componente de varianza“); see also other tables for language corrections
Answer:
Done

Reviewer 3 Report
This is a fairly well-written manuscript that attempts to estimate the heritability of birth weight and weaning weight in the two most prevalent hair sheep breeds in Mexico. The methods are sound and the conclusions are appropriate for the data presented. There is significant concern regarding the narrow scope of the manuscript and its appeal to the larger readership of the journal in this special edition.
Minor comments
There are 2 instances in the Abstract where "weeding" is used in place of weaning.
On line 115, the term "calving" is used instead of lambing.
Stallion is used throughout where ram is the common term for the male parent. Mothers is also used in place of ewes. Suggest either use the terms sire and dam or ram and ewe throughout.
On line 224, it is unclear what the following "Pitono and James [19] with to tropical breed," perhaps it is supposed to be "two tropical breeds"
The Discussion section of the paper is exceptionally hard to follow with all the comparisons between studies. Perhaps a table presenting these historical data would allow the Discussion to become clearer without all the references in the text.
Author Response
Reply to reviewer 3
Thanks for the comments, I also comment the following:
The manuscript fits very well with the scope of the special issue of the journal (Animal Genetics and Genomics). The work presented presents results in which the estimators of direct heritability of two breeds of sheep introduced in Mexico, where there are few studies related to the subject, stand out.
Likewise, I inform you that each of his comments were addressed to substantially improve the work presented. The attached file shows the changes marked in blue.
Minor comments
Reviewer Comments
There are 2 instances in the Abstract where "weeding" is used in place of weaning.
Answer:
The Dramatic error is eliminated, the correct thing is weight at weaning.
Reviewer Comments
On line 115, the term "calving" is used instead of lambing.
Answer:
The Dramatic error is eliminated
Reviewer Comments
Stallion is used throughout where ram is the common term for the male parent. Mothers is also used in place of ewes. Suggest either use the terms sire and dam or ram and ewe throughout. On line 224, it is unclear what the following "Pitono and James [19] with to tropical breed," perhaps it is supposed to be "two tropical breeds"
Answer:
Improved wording as suggested by reviewer.
Reviewer Comments
On line 224, it is unclear what the following "Pitono and James [19] with to tropical breed," perhaps it is supposed to be "two tropical breeds"
Answer:
Improved wording as suggested by reviewer.
Reviewer Comments
The Discussion section of the paper is exceptionally hard to follow with all the comparisons between studies. Perhaps a table presenting these historical data would allow the Discussion to become clearer without all the references in the text.
Answer:
The idea of appending a table seems good to me, however, and verifying the operation rules of the magazine, it is not allowed to use a table in this section, however the discussion of the work is substantially improved.

Round 2
Reviewer 1 Report
Dear Authors,
some issues must still be addressed before publication. A suggestion for your next papers: ask to another “native speaker of the English language (USA)” to check your manuscripts because this one still has several important errors.
What did you mean with “and not because of that, the language is an obstacle that prevents the publication of these results.” ? The poor language is certainly an obstacle for the publication. If the English of the manuscript is really poor, as in this case, the readers will not understand well your analyses and results and, therefore, the manuscript will be useless for the scientific community.
Line-by-line comments
Line 15: old acronyms PC and PP, please change them
Line 91: P75 should be WW as in the other points.
Line 234: “milk production. of the sheep” please remove the full stop
Line 237: “reaches” should be “reached”
Line 238: “conclude” should be “concluded”
Lines 241-246: some verbs are in the present form, while others in the past form. Please fix all the verbs in the past
Line 247: “heritability’s” why this?
Line 258: “but lower than those found in the Blackbelly breed of the present investigation” should be “but lower than those found for the Blackbelly breed in the present investigation”
Line 274: I would not define 0.07 and 0.23 as “moderately heritable”
Line 283: “decisions. suboptimal selection” remove the full stop or remove “suboptimal selection”
Lines 286-288: “in which 126 rams and 964 rams were used. sheep and 3,355 records were analyzed, where they obtained a direct heritability estimator of 0.09 for weaning weight at 60 days of age, lower results with the estimators reported here” check this sentence, “lower results with the estimators reported here” makes no sense. It should be “lower values compared to the ones estimated in the present study” or something like that.
Lines 291-295: “However, the scientific literature also reports the importance of direct heritability factors for weaning weight, in this sense, Analla and Serradilla [20], report results of a study carried out in Spain with the Merino breed, in which were used 126 rams and 964 ewes, and 3,35 records were analyzed, where they obtained a direct heritability estimator of 0.09 for weaning weight at 60 days of age, lower results with the estimators reported here.” Also, this sentence is not quite English. You cannot use “in this sense” in that way; in the passive sentences you must put the verb after the subject: “in which were used 126 rams and 964 ewes” is wrong and it should be “in which 126 rams and 964 ewes were used” or “in which the authors analyzed 126 rams and 964 ewes” if you want an active sentence.
Line 297: “P75” fix this
Line 301: “race” I guess you meant “breed”
Lines 302-305: check this sentence
Line 306: “Similar results in the present investigation for the Pelibuey breed” this sentence has no verb‼
Author Response
I understand the importance of language and I share with You the idea of ​​a simple language for the readers of the scientific community. That is why the work was reviewed by a native speaker of the English language (USA) to comply with this observation.
Regarding the question: What did you mean by "and not because of that, the language is an obstacle that prevents the publication of these results". ?
I mean the following: The journal offers a platform that we can use to improve the review of the language of the work, as long as two or more reviewers suggest it, in this case, only You are pointing it out, however, and if you request it again, we can choose to take this option.
In this sense, we want to say that the language can be solved and improved and that it be not the only criterion of the evaluator to make a decision to reject the results presented.
Likewise, I inform you that each of your comments were met to substantially improve the work presented. In the attached file, the changes marked in yellow color are shown.
Reviewer Comments
Line-by-line comments
Line 15: old acronyms PC and PP, please change them. Answer: Done.
Line 91: P75 should be WW as in the other points. Answer: Done.
Line 234: “milk production. of the sheep” please remove the full stop. Answer: Done.
Line 237: “reaches” should be “reached” Answer: Done.
Line 238: “conclude” should be “concluded” Answer: Done.
Lines 241-246: some verbs are in the present form, while others in the past form. Please fix all the verbs in the past. Answer: Done.
Line 247: “heritability’s” why this? Answer: The writing of the text is improved.
Line 258: “but lower than those found in the Blackbelly breed of the present investigation” should be “but lower than those found for the Blackbelly breed in the present investigation” Answer: Done, reviewer's suggestion accepte.
Line 274: I would not define 0.07 and 0.23 as “moderately heritable” Answer: It is true, it is changed by less inheritable.
Line 283: “decisions. suboptimal selection” remove the full stop or remove “suboptimal selection” Answer: Suboptimal selection removed.
Lines 286-288: “in which 126 rams and 964 rams were used. sheep and 3,355 records were analyzed, where they obtained a direct heritability estimator of 0.09 for weaning weight at 60 days of age, lower results with the estimators reported here” check this sentence, “lower results with the estimators reported here” makes no sense. It should be “lower values compared to the ones estimated in the present study” or something like that. Answer: The writing of the text is improved.
Lines 291-295: “However, the scientific literature also reports the importance of direct heritability factors for weaning weight, in this sense, Analla and Serradilla [20], report results of a study carried out in Spain with the Merino breed, in which were used 126 rams and 964 ewes, and 3,35 records were analyzed, where they obtained a direct heritability estimator of 0.09 for weaning weight at 60 days of age, lower results with the estimators reported here.” Also, this sentence is not quite English. You cannot use “in this sense” in that way; in the passive sentences you must put the verb after the subject: “in which were used 126 rams and 964 ewes” is wrong and it should be “in which 126 rams and 964 ewes were used” or “in which the authors analyzed 126 rams and 964 ewes” if you want an active sentence. Answer: The writing of the text is improved and reviewer's suggestion accepte.
Line 297: “P75” fix this. Answer: Done.
Line 301: “race” I guess you meant “breed” Answer: Done.
Lines 302-305: check this sentence. Answer: The writing of the text is improved.
Line 306: “Similar results in the present investigation for the Pelibuey breed” this sentence has no verb‼ The writing of the text is improved.
